# Nanoresolution real-time 3D orbital tracking for studying mitochondrial trafficking in vertebrate axons in vivo

Fabian Wehnekamp[1†], Gabriela Plucińska[2†‡], Rachel Thong[2], Thomas Misgeld[2*], Don C Lamb[1*]

[1]Department of Chemistry, Center for Nano Science (CENS), Center for Integrated Protein Science (CIPSM) and Nanosystems Initiative München (NIM), Ludwig Maximilians-Universität München, Munich, Germany; [2]Munich Cluster for Systems Neurology (SNergy), Center for Integrated Protein Science (CIPSM), German Center for Neurodegenerative Diseases (DZNE), Institute of Neuronal Cell Biology, Technische Universität München, Munich, Germany

**Abstract** We present the development and in vivo application of a feedback-based tracking microscope to follow individual mitochondria in sensory neurons of zebrafish larvae with nanometer precision and millisecond temporal resolution. By combining various technical improvements, we tracked individual mitochondria with unprecedented spatiotemporal resolution over distances of >100 μm. Using these nanoscopic trajectory data, we discriminated five motional states: a fast and a slow directional motion state in both the anterograde and retrograde directions and a stationary state. The transition pattern revealed that, after a pause, mitochondria predominantly persist in the original direction of travel, while transient changes of direction often exhibited longer pauses. Moreover, mitochondria in the vicinity of a second, stationary mitochondria displayed an increased probability to pause. The capability of following and optically manipulating a single organelle with high spatiotemporal resolution in a living organism offers a new approach to elucidating their function in its complete physiological context.
DOI: https://doi.org/10.7554/eLife.46059.001

**\*For correspondence:**
thomas.misgeld@tum.de (TM);
d.lamb@lmu.de (DCL)

[†]These authors contributed equally to this work

**Present address:** [‡]Cell Biology, Department of Biology, Faculty of Science, Utrecht University, Utrecht, The Netherlands

**Competing interests:** The authors declare that no competing interests exist.

## Introduction

Neurons critically depend on proper positioning of organelles along their axons. While the basic molecular players of the axonal transport machinery are well established (*Hirokawa et al., 2010*), the higher order levels of regulation that govern the overall 'life cycle' of axonal organelles are poorly understood (*Misgeld and Schwarz, 2017*; *Plucińska and Misgeld, 2016*). One example of this dearth of understanding is provided by mitochondria where their proper distribution and turnover are of seminal importance for neuronal homeostasis (*Chang and Reynolds, 2006*; *Sheng and Cai, 2012*; *MacAskill and Kittler, 2010*; *Saxton and Hollenbeck, 2012*). Despite the well-established roles of specific kinesin and dynein motors in the long-distance transport of axonal mitochondria and some recent progress on the mechanisms of their local anchorage and degradation (*MacAskill and Kittler, 2010*; *Saxton and Hollenbeck, 2012*; *Sheng, 2014*; *Ashrafi and Schwarz, 2013*), many details regarding how the distribution of how mitochondria are established, maintained and regulated – especially in vivo – remain elusive. For example, the regulatory interplay of molecular motors that propel and brake mitochondria and give rise to their characteristic 'saltatory' movement (*Morris and Hollenbeck, 1993*) and additional 'non-canonical' movement behaviors (*Chang and Reynolds, 2006*; *Morris and Hollenbeck, 1993*; *Ligon and Steward, 2000*) is not well understood. Our understanding of the origin, travel range and final destination of transported

mitochondria is only emerging (*Misgeld et al., 2007*; *O'Toole et al., 2008*), and how the local cellular microenvironment, such as activity or calcium levels, influences mitochondrial motility in axons with a realistic in vivo geometry and surrounding is only now starting to be explored (*Faits et al., 2016*; *Sajic et al., 2013*; *Ohno et al., 2011*; *Smit-Rigter et al., 2016*; *Lewis et al., 2016*).

In many respects, the gap in understanding between the well-established molecular underpinnings of mitochondrial transport gleaned from biophysical studies in vitro and the bigger picture of this organelle's homeostasis in neurons in vivo can be attributed to a lack of techniques that can span the different spatial domains involved: Motors step on the scale of nanometers but propel organelles on the scale of many hundreds of micrometers. Here, we present an approach based on 3D single particle tracking (3D SPT) that is suitable to bridge this chasm and demonstrate the in vivo applicability of this tool to the zebrafish model of axonal transport (*O'Donnell et al., 2013*; *Plucińska et al., 2012*). This new approach will be instrumental in linking the biophysical understanding of single organelle dynamics to physiological and pathological neuronal changes in vivo.

In the past few years, a number of 3D SPT methods have been developed. They rely on different approaches like altering the shape of the point spread function (*Spille et al., 2015*; *Kao and Verkman, 1994*; *Shechtman et al., 2015*) or 'lock-in' feedback loops that re-center the laser focus onto the tracked particle (*Welsher and Yang, 2014*; *Dupont et al., 2013*; *McHale et al., 2007*; *Perillo et al., 2015*; *Juette and Bewersdorf, 2010*; *Levi et al., 2005*). These approaches have established 3D SPT as a powerful tool to study the dynamics of subcellular structures. However, these techniques have largely remained restricted to cells in dissociated culture and other reductionist models due to technical limitations. To successfully track individual mitochondria inside an intact organism, such as a live zebrafish larva, several points have to be considered. First, due to fusion and fission events, the shape of each mitochondrion is changing during the transport along the neuron and thus the tracking technique must be resilient to such shape changes. Second, transport in axons, even in a small organism like a fish larva, potentially extends over distances of hundreds of micrometers. Thus, the available tracking range has to be expanded well beyond a single field of view. Third, to achieve prolonged trajectories of single moving mitochondria against a dense backdrop of resting organelles, the fluorescence of each mitochondrion has to be individually controlled. Finally, the recorded trajectories have to be contextualized by vistas of the cellular environment in which the organelle traveled.

Here, we present a 3D SPT method based on real-time 3D orbital tracking that is able to overcome these challenges: The method combines high spatial (XY:<5 nm Z:<30 nm) and temporal resolution (100 Hz) with simultaneous wide-field imaging and local photo-activation. A feedback loop recurrently re-centers the specimen on top of the microscope once a tracked particle approaches the edge of the field of view. In combination with previously developed genetic tools (*Plucińska et al., 2012*), we are capable of tracking single neuronal mitochondria in vivo over distances of more than 100 μm. The exquisite spatiotemporal resolution of the microscope system allowed us to discriminate not only the canonical fast components during active motion, but also revealed a previously undetected motional state in both the antero- and retrograde directions. This state has a slower velocity and was engaged when mitochondria undergo temporary directional changes. A detailed examination of transitions between motion states and pause durations showed a complex pattern governing such transient changes in mitochondrial motility. The combination of trajectory and 'environmental' data (i.e. wide-field images of the region surrounding the tracked particle) allowed us to analyze the influence of other mitochondria present in the axon and showed that stationary mitochondria can act as roadblocks that initiate the slower motional state in passing organelles, a potential mechanism for overcoming such physiological obstacles.

## Results

### In vivo 3D orbital tracking in zebrafish

To track organelles with nanometer precision and millisecond temporal resolution in zebrafish larvae, we further developed the 3D orbital SPT microscope described previously (*Dupont et al., 2013*; *Katayama et al., 2009*) (*Figure 1*) to overcome four essential limitations for 3D in vivo SPT applications: (I) We obtained precise hardware synchronization using a field programable gate array as well as sub millisecond timing of the tracking feedback loop algorithm by executing the algorithm on a

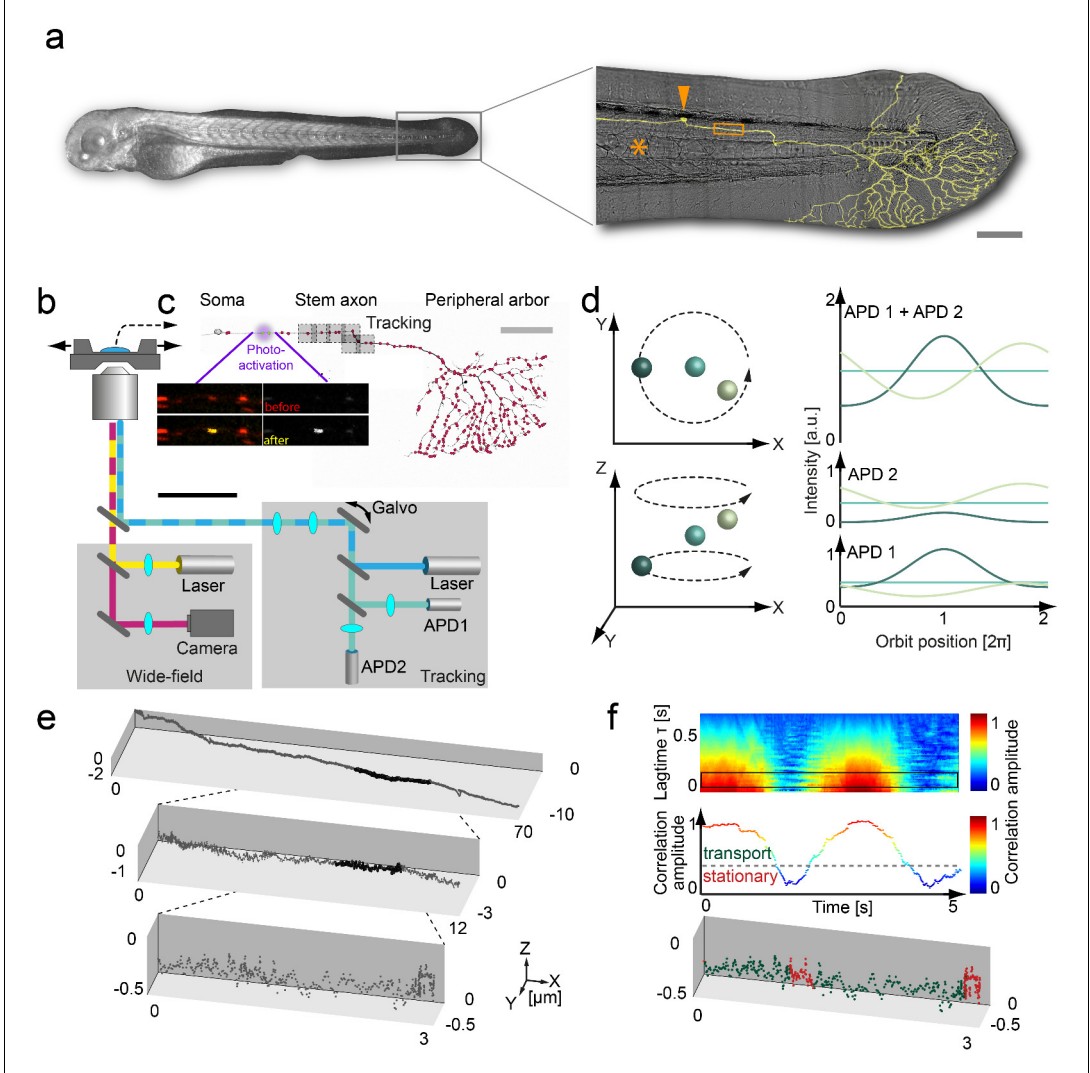

**Figure 1.** 3D orbital tracking microscope and mitochondrial trajectory analysis. (**a**) Light microscopy transmission image of the zebrafish and a zoom in on the tail with a typical Rohon-Beard neuron labeled by a membrane-targeted fluorescent protein (shown in yellow). The typical tracking area (orange box), soma (orange arrow) and notochord (orange asterisks) are indicated to provide a contextual overview (scale bar, 200 µm). (**b**) Schematic of the custom-built 3D real-time orbital tracking microscope consisting of a confocal tracking channel and a wide-field channel for simultaneous environmental observation. (**c**) A confocal reconstruction of a sensory neuron is shown where both the membrane and the individual mitochondria, indicated schematically as red points, are fluorescently labeled (scale bar, 100 µm). The imaging sites in the stem axon are shown in gray with the multiple boxes indicating the re-location of the field of view during long-range tracking. Images of TagRFP (in red)/PA-GFP-labeled (in yellow) axonal mitochondria before (upper image pair) and after (lower image pair) photo-activation of a single mitochondrion are shown (scale bar, 5 µm). (**d**) Schematic representation of the 3D orbital tracking approach. Different particle locations are indicated through spheres of varying color. Lateral localization is performed by orbiting the laser focus around the particle of interest (top left). The amplitude and peak position of the intensity orbit depends on the position of the particle in relation to the center of the orbit (top right). The color of the line represents the signal coming from the object of the corresponding color in the left panels. The axial localization is achieved by using two confocal detection volumes placed equidistant above and below the focal plane (bottom left), so the intensity ratio between the two planes is proportional to the axial position of the particle (bottom right). (**e**) A trajectory of an anterograde moving mitochondrion (100 Hz, 20,000 data points). Zoom-ins illustrate the actual density of the acquired data points. (**f**) Autocorrelation carpet (top) of the angle between consecutive orbits. The black box indicates the lag time τ region averaged for the plot shown in the middle panel. Dashed line marks the threshold that was used to separate stationary phases (red points in bottom panel) from directed motion (green data points) in the lower plot. The lower plot is the same as shown in the maximum zoom-in in panel e). Galvo: galvanometer mirrors; APD, avalanche photodiode.

DOI: https://doi.org/10.7554/eLife.46059.002

The following source data and figure supplements are available for figure 1:

**Source data 1.** Matlab files for analyzing and producing graphics of trajectory from panel e and correlation analysis from panel f .

*Figure 1 continued on next page*

*Figure 1 continued*

DOI: https://doi.org/10.7554/eLife.46059.009

**Figure supplement 1.** | Orbital tracking precision.

DOI: https://doi.org/10.7554/eLife.46059.003

**Figure supplement 1—source data 1.** Matlab and data files for analyzing and producing graphics for particle localization from panels a-f.

DOI: https://doi.org/10.7554/eLife.46059.004

**Figure supplement 2.** Heartbeat of the larvae.

DOI: https://doi.org/10.7554/eLife.46059.005

**Figure supplement 2—source data 1.** Matlab files for analyzing and producing graphics of the heartbeat analyis in panels a-d.

DOI: https://doi.org/10.7554/eLife.46059.006

**Figure supplement 3.** Influence of mitochondrial shape on localization precision.

DOI: https://doi.org/10.7554/eLife.46059.007

**Figure supplement 3—source data 1.** Matlab files for producing graphics of size dependent localization precision of mitochondria from panel b.

DOI: https://doi.org/10.7554/eLife.46059.008

real-time operating system. (II) We increased the lateral tracking range to centimeters by amending the 3D orbital tracking algorithm with a long-range tracking feature that automatically re-centers the sample stage before the particle exits the field-of-view. (III) We incorporated multiple laser lines into the orbit light path – including a 405 nm laser for controlled photo-activation, which allows single organelle tracking in a densely labeled background. (IV) To decrease photobleaching and phototoxicity, we implemented dark orbits, where only a subset of orbits is illuminated by the excitation laser, which significantly extends experiment duration in exchange for a moderate decrease in temporal resolution. This combination of technical advances makes it possible to track single organelles in a living organism in 3D with nanometer scale spatial resolution, millisecond scale temporal resolution and over lateral distances of centimeters.

We applied orbital tracking to a previously published zebrafish model (*Plucińska et al., 2012*). To do so, we co-expressed mitochondrially targeted red fluorescent protein (mitoTagRFP-T) and photo-activatable green fluorescent protein (mitoPAGFP) in sensory (Rohon-Beard) neurons using the Gal4/UAS

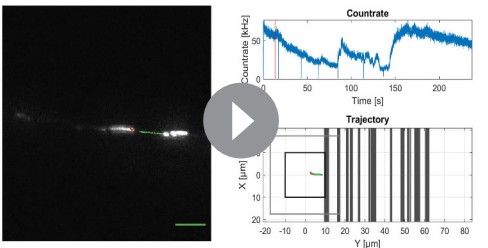

**Video 1.** Anterograde transport. (Left) Wide-field video showing the anterograde transport of a photo-activated mitochondrion along a single axon. Color-coding of the trailing points indicate different movement states (green – fast anterograde, yellow – slow anterograde, orange – slow retrograde, red – stationary state; scale bar 5 μm). (Top right) Mean photon count rate of both detection channels. Downward spikes indicate long range tracking events where the tracking software is not able to track the particle for ~35–70 ms (axis dependent). The laser intensity was occasionally increased manually to ensure high tracking accuracy. (Bottom right) 3D trajectory of the moving mitochondrion. The field of view of the EMCCD camera is indicated by the gray square box, the threshold for the long-range tracking in black. Gray vertical lines indicate the position of stationary mitochondrion. After 145 s, the tracking algorithm switches to a brighter, stationary mitochondrion.

DOI: https://doi.org/10.7554/eLife.46059.010

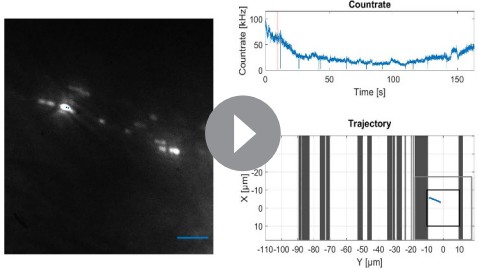

**Video 2.** Retrograde transport. (Left) Wide-field video showing the retrograde transport of a photo-activated mitochondrion along a single axon. Color-coding of the trailing points indicate different movement states (blue – fast retrograde, orange – slow retrograde, yellow – slow anterograde, red – stationary state; scale bar 5 μm). (Top right) Mean photon count rate of both detection channels. Downward spikes indicate long range tracking events where the tracking software is not able to track the particle for ~35–70 ms (axis dependent).

DOI: https://doi.org/10.7554/eLife.46059.011

system (see Materials and methods). At 3 days post fertilization (dpf), we photo-activated a moving mitochondrion in the axon near the soma with a confined *xy* scan of the 405 nm laser beam (*Figure 1a–c*), which enabled tracking of the targeted (photo-converted) organelle unambiguously in the GFP channel against the backdrop of the numerous (non-photo-converted) mitochondria visible in the RFP channel. Trajectories of moving mitochondria were derived from recordings of orbit and stage displacements needed to follow the organelle through the axon. A wide-field view of the tracked mitochondria was recorded in parallel. The combined technical improvements to expand tracking range and reduce phototoxicity allowed us to acquire mitochondrial trajectories of >100 μm with ~5 nm precision using a 5 ms orbit (*Figure 1d,e,SI*, *Figure 1—figure supplement 1*, *Video 1* and *2*). Given the extremely high spatiotemporal resolution of the data, different precautions had to be made. For example, stem axons nearby blood vessels exhibited low-amplitude fluctuations perpendicular to mitochondrial movement due to the heartbeat (*Figure 1—figure supplement 2*). This artifact could be minimized by choice of the recording site. Trajectories that were significantly impacted by blood flow could be easily identified and discarded based on the amplitude and frequency characteristics of the heartbeat. We also verified that the shape of the mitochondria did not impact our tracking precision (Material and methods and *Figure 1—figure supplement 3*).

## Two types of motion drive mitochondrial transport

To differentiate between active phases of transport along the axon and stationary phases (pauses), we used an autocorrelation analysis of the angle between two consecutive localizations in the trajectory (*Figure 1f*). Within a window of 64 data points, the autocorrelation amplitude for active phases approached a value of one, while, in the stationary state, this value dropped below 0.30 (a threshold value generated from randomization of the individual trajectory data, see Materials and methods). The stationary state does not imply that the motors are not intact or that the mitochondria are immobile, only that no net progress is observed in any particular direction over a time window of hundreds of milliseconds. Indeed, from conventional time-lapse imaging experiments, different behaviors of stationary mitochondria are also known (such as a local 'wiggling' as opposed to a fully immobile state, *Misgeld et al., 2007* – which likely represent different modes or extents of anchorage; *Gutnick et al., 2019*; however, here we focused on the translocation behavior of mitochondria).

Analysis of the 3D trajectories revealed that antero- and retrograde-directed phases of active movement were composed of two distinct fast and slow movement states, which differed significantly in speed and processivity (*Figure 2*, *Figure 2—figure supplement 1*, *Table 1*). To validate the measured values, we compared the average velocities of motion in the anterograde and retrograde direction with results of a previous study (*Plucińska et al., 2012*). The obtained values of the 3D Orbital tracking approach are consistent with the previous study when accounting for the difference in time resolution (10 ms versus 500 ms, leading to the inability to properly discriminate between fast and slow components in the previous study) and temperature (25°C here versus 28°C in *Plucińska et al., 2012*, *Table 2*). Fast states are responsible for long-distance trafficking of the mitochondria with average velocities of 0.62 μm/s for anterograde motion and 0.72 for retrograde motion. The slow states carry out movements over shorter distances and shorter timescales (*Figure 2c–f*). The characteristics of the slow movement states (i.e. duration, displacement and velocity) are similar in both directions (*Figure 2c*). Although the average travel distances and times for the slow movement states are small, the orbit displacement (*Figure 2c*) and velocity histograms (*Figure 2—figure supplement 1a,b*) clearly indicate directional motion, which demonstrates a proper separation of these processes from the stationary state.

Mitochondria typically showed sustained phases of fast antero- or retrograde motion, which were interspersed by short-term pauses and periods of slow directed motion where the direction of motion can reverse. After a short period of time, the mitochondria continued to travel fast in the original direction of transport. This stereotypical sequence of events suggests a high level of coordination between the motion states. To quantify this, we analyzed the frequency and transition time between the different states (*Figure 2—figure supplement 2*). A number of trends became apparent from this analysis. First, an analysis of the pause durations suggests two different pausing mechanisms: Pauses involving fast motion states (fast-fast or fast-slow/slow-fast transitions) show monoexponential distributions with decay constants of 1.94 s and 1.97 s, whereas pauses between two slow states show a longer duration with a decay constant of 3.2 s (*Figure 2—figure supplement 2*).

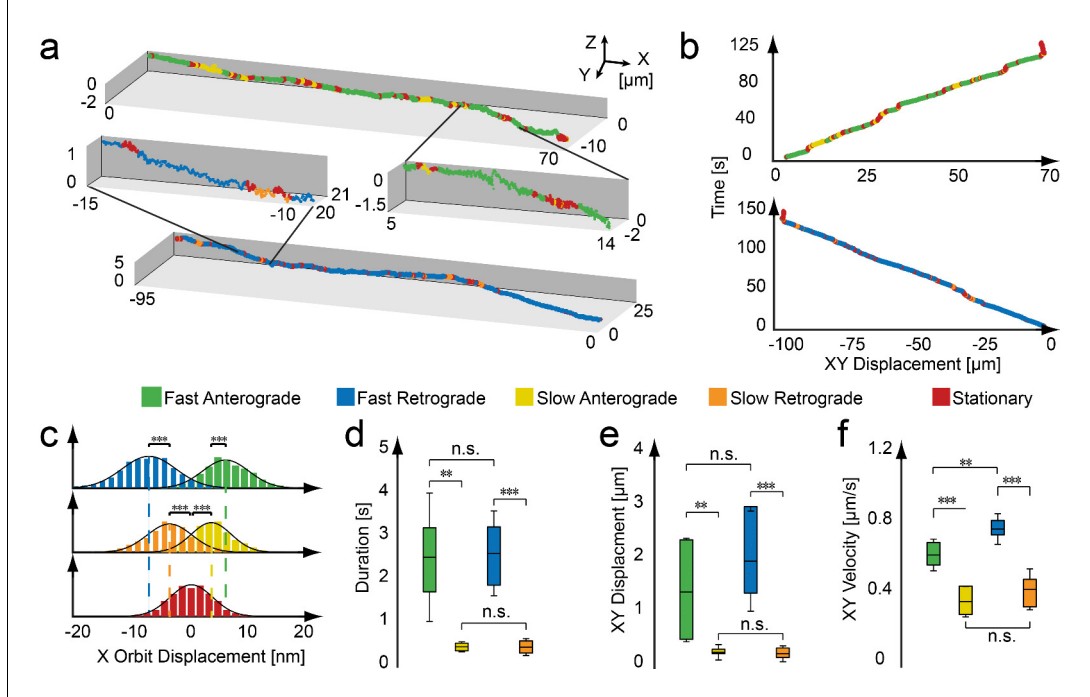

**Figure 2.** Active and stationary states of mitochondrial movement. (a) Representation of an anterograde (**top**) and retrograde (**bottom**) trajectory. Color coding indicates phases of fast motion (green - anterograde; blue - retrograde), slow motion (yellow - anterograde; orange - retrograde) and stationary phases (red). (b) Kymographs of the trajectories shown in panel a. (c–f) Population properties determined from 43 mitochondrial trajectories collected from 16 embryos. (c) Orbit displacement histograms for the anterograde direction (fast: 5.8 ± 5.0 nm, n = 83,272 orbits; slow: 3.4 ± 3.7 nm, n = 27,638), the retrograde direction (fast: −7.1 ± 5.3 nm, n = 61,217; slow: −3.7 ± 4.4 nm, n = 21,984) and the stationary states (0.0 ± 3.9 nm, n = 1,153,949). Dashed lines indicate the center of the Gaussian distributions. (d) Durations of anterograde (fast: 2.5 ± 1.5 s; n = 331 states; slow 0.46 ± 0.11 s; n = 416) and retrograde motion states (fast: 2.6 ± 1.0 s; n = 220; slow 0.45 ± 0.20 s; n = 339). (e) XY displacement during anterograde (fast: 1.5 ± 1.0 μm; n = 331; slow 0.30 ± 0.15 μm; n = 416) and retrograde motion states (fast: 2.1 ± 1.0 μm; n = 220; slow 0.27 ± 0.15 μm; n = 339). (f) Lateral velocity during anterograde (fast: 0.62 ± 0.09 μm/s; n = 331; slow 0.36 ± 0.08 μm/s; n = 416) and retrograde states (fast: 0.76 ± 0.08 μm/s; n = 331; slow 0.42 ± 0.11 μm/s; n = 416). Box plot shows the average, as wells as 25 and 75 percentile; error bars indicate standard deviation. Asterisks indicate significance levels (determined by a two-sided t-test) of *: $p < 0.01$, **: $p < 0.005$ and ***: $p < 0.001$. (**Table 1**).

DOI: https://doi.org/10.7554/eLife.46059.012

The following source data and figure supplements are available for figure 2:

**Source data 1.** Matlab files for analyzing and producing graphics of trajectories from panel a, kymographs from panel b, and population properties from panels c-f.

DOI: https://doi.org/10.7554/eLife.46059.017

**Figure supplement 1.** Assignment of transport populations using a maximum likelihood approach.

DOI: https://doi.org/10.7554/eLife.46059.013

**Figure supplement 1—source data 1.** Matlab files for analyzing and producing graphics of velocity distribution plots from panels a and b as well as corresponding population assignment from panels c and d.

DOI: https://doi.org/10.7554/eLife.46059.014

**Figure supplement 2.** Transition pattern of mitochondria between different movement phases.

DOI: https://doi.org/10.7554/eLife.46059.015

**Figure supplement 2—source data 1.** Matlab files for producing graphics of time decay plots.

DOI: https://doi.org/10.7554/eLife.46059.016

Very long pauses (>20 s) were always (98% of n = 97) associated with transitions involving the slow motion states and had a high probability of involving a change in directionality (56% of n = 97). For pauses <20 s, a change in direction was only detected in 19% (of n = 1137) of the events. Second, there is a clear directionality of motion in the overall trajectory determined by the direction of fast motion. We did not observe any transitions between fast anterograde and fast retrograde states within one track, not even multi-step transitions that would involve the slow state. Moreover, moving mitochondria – irrespective of their speed – have a high probability of continuing in the same

**Table 1.** Dynamic states properties of untreated zebrafish embryos.

Numerical values and statistics of zebrafish embryos for data shown in *Figure 2c–f*. Values are gives as the average ±s.d. Significance levels were determined using a two-sided t-test). The left and right cells highlighted by the gray boxes indicate the value pair used to determine the respective p values.

| | Fast anterograde | Slow anterograde | Fast retrograde | Slow retrograde | Stationary |
|---|---|---|---|---|---|
| Duration | 2.53 ± 1.48 [s] | 0.46 ± 0.11 [s] | 2.62 ± 0.98 [s] | 0.45 ± 0.20 [s] | |
| n | 331 | 416 | 220 | 339 | |
| p | 1.7e-3 | | 5.1e-5 | | |
| p | | 0.88 | | | |
| p | | | 0.92 | | |
| XY Displacement | 1.47 ± 0.96 [µm] | 0.30 ± 0.15 [µm] | 2.07 ± 0.97 [µm] | 0.27 ± 0.15 [µm] | |
| n | 331 | 416 | 220 | 339 | |
| p | 3.9e-3 | | 2.2e-4 | | |
| p | | 0.18 | | | |
| p | | | 0.67 | | |
| XY Velocity | 0.62 ± 0.09 [µm/s] | 0.36 ± 0.08 [µm/s] | 0.76 ± 0.08 [µm/s] | 0.42 ± 0.11 [µm/s] | |
| n | 331 | 416 | 220 | 339 | |
| p | 2.7e-6 | | 1.0e-6 | | |
| p | | 1.6e-3 | | | |
| p | | | 0.14 | | |
| X Orbit Displacement | 5.8 ± 5.0 [nm] | 3.4 ± 3.7 [nm] | −7.1 ± 5.3 [nm] | −3.7 ± 4.4 [nm] | 0.0 ± 3.9 [nm] |
| n | 83272 | 27638 | 61217 | 21984 | 1153949 |
| p | 2.8e-5 | | 1.3e-7 | | |
| p | | 6.1e-7 | | | |
| p | | | | 5.3e-7 | |
| p | | 0.49 | | | |

DOI: https://doi.org/10.7554/eLife.46059.018

direction after a pause (77% or higher depending on the mode of motion, *Figure 2—figure supplement 2*). Approximately half of the time, the mitochondria remain in the same motional state (i.e. direction and velocity) and directional changes between fast and slow motion were unlikely (~6% of all pauses). In summary, the probability diagram of possible transitions (*Figure 2—figure supplement 2*) is substantially more complex than anticipated from previous in vitro and in vivo reports (*Misgeld et al., 2007*; *Plucińska et al., 2012*; *Obashi and Okabe, 2013*) and suggests that a number of mechanisms might influence the different state transitions along a mitochondrion's axonal trajectory.

## Influence of obstacles

Utilizing the wide-field information that complements our trajectories, we analyzed the dynamic behavior of moving mitochondria in the context of the surrounding axonal environment (*Figure 3a*). Here, the influence of stationary mitochondrion became evident, as suggested previously (*Ohno et al., 2011*; *Figure 3a*, *Video 1* and *2*). Mitochondria moving along a 'free' track spend the majority of time in fast movement states (*Figure 3b*). However, when stationary mitochondria were present on the track, the time spent in the stationary state substantially increased in both directions (anterograde: 34% to 58%, retrograde: 40% to 58%). Notably, for anterograde transport, the time spent in the fast movement state decreased substantially at sites occupied by resting mitochondria (ratio fast/slow movement states: free track - 56%/10% = 5.6 vs. occupied track - 27%/14% = 1.9). For transport in the retrograde direction, this switch was absent (free track - 52%/8% = 6.5 vs. occupied track - 37%/4% = 9.3). This is in line with previous observations that retrograde transport might

**Table 2.** Velocity comparison.

Comparison of velocities between the orbital tracking analysis and the wide-field analysis used in a previous study (*Plucińska et al., 2012*). The comparison between the wide-field analyses at 25°C and 28°C showed a 40% reduction in velocity, which is attributed to the reduced temperature. Due to the low time and spatial resolution, the wide-field analysis can only extract the velocity for a single population. This velocity value represents an average of the fast, slow and short stationary states, due to the inability of the wide-field analysis to reliably discriminate between these states. Values are given as the average ±s.d.

**Lateral velocity**

| Population | Tracking analysis 25°C | Wide-field recording analysis 25°C | Wide-field analysis (after Plucinska et al.) 28°C |
|---|---|---|---|
| Retrograde | | | |
| Fast | 0.76 ± 0.08 [μm/s] | 0.55 ± 0.07 [μm/s] | 0.92 ± 0.02 [μm/s] |
| Slow | 0.42 ± 0.11 [μm/s] | | |
| Anterograde | | | |
| Fast | 0.62 ± 0.09 [μm/s] | 0.45 ± 0.08 [μm/s] | 0.77 ± 0.01 [μm/s] |
| Slow | 0.36 ± 0.08 [μm/s] | | |

DOI: https://doi.org/10.7554/eLife.46059.019

be less sensitive towards obstacles than anterograde transport (*Mallik et al., 2004*), suggesting that the slow movement state might be 'a shift to low gear' used to circumvent obstacles that the driving forces of the fast transport states have a hard time overcoming.

It is known that road-blocks or local influences exist along the axon that may provide external determinants of pausing (*Hirokawa et al., 2009*; *Conde and Cáceres, 2009*; *Bálint et al., 2013*). To explore the possibility of preferred pausing sites and check whether pausing is localized at specific points or randomly distributed along the axon, we took advantage of the fact that, with a slight modification of our approach, we can observe the same axon segment with several distinct cargoes (rather than following one cargo over an extended axon length). For these measurements, we used a combination of mitoTagRFP-T and mitoDendra2. This allowed us to create a 'red only' field-of-view by taking advantage of the swift conversion of Dendra2's green to a red state using 405 nm excitation (*Chudakov et al., 2007*). In this field-of-view, the incoming mitochondria appear green and were easily tracked. We acquired trajectories of 16 mitochondria along the same region. When we plotted the time needed to advance 100 nm against the position of the repeatedly sampled axon stretch, a clear pattern emerged (*Figure 3c*): While progress along most of the axonal length was relatively steady (t < 1 s for traveling 100 nm), there were distinct foci where progress was slower. Incorporating wide-field information into the analysis, we observed that, in many instances, the pauses coincided as expected with the presence of stationary mitochondria. However, other pause foci appear on 'free' segments – suggesting that there are additional retaining influences that can induce pausing, and that the clustering of pauses near stationary mitochondria might not only be merely due to geometrical constraints, but could be due to the local structure of the cytoskeleton (*Bálint et al., 2013*) or attractive milieu influences, such as local calcium hot spots (*MacAskill and Kittler, 2010*) or areas of substrate availability (*Pekkurnaz et al., 2014*).

## Discussion

In summary, this new application of orbital tracking microscopy with nanoscopic spatial precision and millisecond temporal resolution to zebrafish neurons in vivo provides unprecedented detail of subcellular trafficking in an intact organismic context. While this is the first time a feedback-based 3D SPT approach was used for tracking particles in vivo, we want to point out that this is not the only technique capable of such measurements (*Shechtman et al., 2015*; *McHale et al., 2007*), and further improvements are conceivable. For example, two-photon excitation-based tracking might in principle be more favorable for deep tissue imaging than our one-photon approach (*Helmchen and Denk, 2005*). Still, the method presented here is less costly and – given the broad two-photon absorption cross-section of many fluorophores – more versatile as far as wavelength multiplexing is concerned. Moreover, the specific implementation of 3D SPT that we detail here overcomes field-of-

view limitations of many other approaches, allows flexible integration of photo-activation lasers and easy modulation of the tracking laser, which together can further increase the tracking range. Finally, simultaneous wide-field detection provides cellular and histological context, which is important in interpreting local signaling that might affect organelle trafficking.

Capitalizing on these technical advances, we reveal that beyond the expected three major movement states of mitochondria (resting, antero- and retrograde) previously reported in zebrafish and many other settings (*Plucińska and Misgeld, 2016*; *Saxton and Hollenbeck, 2012*), we observed two components, a fast and a slow transport process in both the antero- and retrograde directions. The fast components are consistent with the known characteristics of classical kinesin- and dynein-mediated transport reported in vivo (*Misgeld and Schwarz, 2017*), and – even though the dynamics of organelles result from complex motor combinations and interactions – are on the order of speeds measured for these motors in reconstitution assays (0.7–1.0 µm/s; *Toba et al., 2006*; *King and Schroer, 2000*; *Milo and Phillips, 2016*). While we do not currently know the molecular underpinnings of these slow movement states, our observations reveal a number of properties that deserve note: The slow movement states appeared – in contrast to the fast movements – symmetrical in

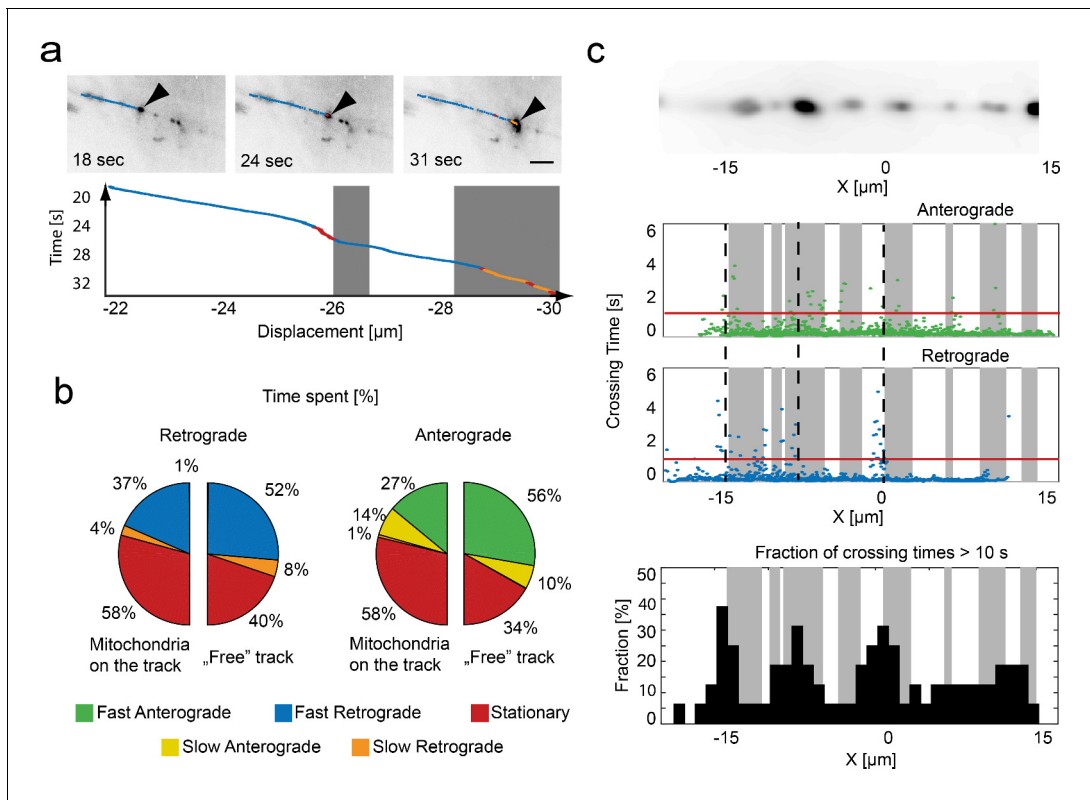

**Figure 3.** Relation of mitochondrial trajectory to local axonal environment. (a) Mapping of the trajectory of a single mitochondrion (black arrow) onto the inverted wide-field images (scale bar, 5 µm). Bottom panel, kymograph color coded according to motion state, location of stationary mitochondria depicted in gray. (b) Pie charts indicating the fraction of time spent in each motion state related to the local presence or absence of a mitochondrion in retrograde (left) or anterograde (right) direction (n = 16 trajectories, nine fish). (c) Repetitive tracking of mitochondria over the same stretch of an axon. *Upper Panel*: Wide field image of the ROI showing the location of stalled mitochondria on a section of microtubules. *Middle panel*: The time a mitochondrion needed to transverse 100 nm is plotted as a function of position along the axon. Gray boxes indicate the presence of stationary mitochondria. The red line indicates the threshold level to identify bins of slow movement. Dashed black lines indicate locations where multiple mitochondria were observed to pause (see lower panel). *Lower panel*: Fraction of trajectories (plotted in 1 µm bins from mitochondria moving in both directions) along the axon showing crossing times of more than 10 s for 1 µm.

DOI: https://doi.org/10.7554/eLife.46059.020

The following source data is available for figure 3:

**Source data 1.** Matlab files for analyzing and producing graphics of pie charts from panel b, crossing time of stationary mitochondria as well as fracrion of trajectories from panel c.

DOI: https://doi.org/10.7554/eLife.46059.021

speed. Moreover, the transition into and out of these slow movement states was not random, but followed a defined set of rules. Finally, these state transitions seemed to be impacted by external influences, such as local obstacles within the axon. Notably, atthe typical sampling rates used in kymography recordings of mitochondrial transport in vitro or in vivo (i.e. 1–2 Hz), the slow state is on the verge of detectability (duration ~0.45 s; *Table 2*) – and hence might sometimes be subsumed in either the moving speeds or convolved with pause length and frequency. There are a number of possible interpretations of the slow movement states. One would be shape changes of mitochondria as they encounter obstacles or are partially tethered in the axon. For instance, one end of the mitochondria could encounter an obstacle and stop moving while the other end continues to move. As the orbital tracking measures the position of the center of fluorescence signal, compression or expansion of the mitochondria would convert into a slower change in the center position. However, this explanation is unlikely as the average displacement in the slow movement state is approximately 300 nm, which would be equivalent to a 600 nm change in the overall shape of the mitochondria. This is on the order of the size of moving mitochondria (*Figure 1—figure supplement 3b*). In addition, we use the presence of a stationary state to separate regions of active transport. In the current implementation of the analysis, we would not be able to detect a direct switch between the fast and slower motional states. If the slow movement state were due to anchoring of one end, it would mean that the entire mitochondria first stops moving and then only one side continues. Still, we would like to note that the orbital tracking method also has the potential to investigate shapes changes during tracking by analyzing higher order Fourier components of the orbital tracking signal or by adding a second, small amplitude higher frequency oscillation to the orbit (*Lanzano et al., 2011*). Thus, a more formal investigation of possible shape changes of mitochondria during different behaviors will become possible in the future.

Given our present data, more likely explanations of the slow movement states in our view involve changes in the engaged molecular transport machinery. Options include: (I) the engagement of additional motors beyond kinesins and dyneins (the known speed characteristics of which matched the fast movement states, as expected – see *Table 1*), (II) the concomitant engagement of an anchor (such as syntaphilin) and kinesin/dynein motors, or (III) the existence of an unappreciated regulation that can switch the known mitochondrial motors into specific 'slow' states. The fact that the slow states had the same speed distribution in both directions leads us to favor the first hypothesis; however, the nature of the involved motors remain elusive as of now. One possibility would be the involvement of actin-based motors in mitochondrial movements, as previously suggested for example based on pharmacological experiments in vitro and genetically in flies (*Ligon and Steward, 2000*; *Morris and Hollenbeck, 1995*; *Pathak et al., 2010*). Thus far, myosin-actin interactions have been mostly linked to reduced mitochondrial motility (*Pathak et al., 2010*) and specific stationary states, as recently demonstrated using a new opto-chemical crosslinking approach (*Gutnick et al., 2019*). Still, our data show a complex and orchestrated sequence of motility changes at transitions from fast moving into stationary states that, in many prior analyses, would go undetected and hence be subsumed in other movement states. Thus, a role for a transient slow actin-dependent movement phase that at the same time is compatible with the residual slow motility of mitochondria in microtubule-depleted axons in vitro (*Morris and Hollenbeck, 1995*), as well as the acceleration of transport seen after myosin knock-down in flies (*Pathak et al., 2010*), seems like a plausible explanation of our observations. The slow movement state engaged preferentially when a moving mitochondrion encountered a local obstacle such as an anchored mitochondrion in the axon shaft. This could be an indication that these putative actin-dependent mitochondria translocations are important for local positioning and movement between microtubule tracks (*Gutnick et al., 2019*; *Atkinson et al., 1992*; *Langford, 1995*) and the notion that myosin-actin interactions increase mitochondrial pause frequencies (*Morris and Hollenbeck, 1995*; *Pathak et al., 2010*). While this specific molecular interpretation remains speculative, the convenience of gene overexpression and deletion in zebrafish combined with 3D orbital tracking now allows testing of this hypothesis.

In any case, while the exact molecular nature and role of the slow movement phases of mitochondrial trafficking remain to be resolved, the observation of new movement states for a well-studied organelle such as mitochondria testifies to the power of the in vivo 3D SPT approach. Indeed, we foresee a range of applications, including investigations using neurological disease models, such as tauopathies, where transitions of movement states and hence the duty cycle between movement and stationary phases seem to be especially affected (*Plucińska et al., 2012*; *Fatouros et al., 2012*;

*Devireddy et al., 2015*). Possible applications of the technique described here go well beyond simple in vivo observations of physiological and pathological organelle behavior. For instance, the ability to visualize the cellular surroundings of a trafficking organelle will allow a detailed determination of how neuronal landmarks known to be preferentially associated with mitochondria, such as branch points (*Faits et al., 2016*; *Courchet et al., 2013*; *Spillane et al., 2013*) or presynaptic terminals (*Lewis et al., 2016*; *Obashi and Okabe, 2013*), modulate mitochondrial behavior. This approach can be readily combined with a rapidly expanding set of organelle-targeted biosensors (*Breckwoldt et al., 2014*) and optogenetic actuators of organelle physiology (*Rost et al., 2015*; *Ashrafi et al., 2014*), as well as motor (*Gutnick et al., 2019*; *van Bergeijk et al., 2015*) and track (*Borowiak et al., 2015*) composition. As the laser beam is orbited about a single organelle, it provides optical selectively and will enable a detailed correlation between the physiological state of a trafficking organelle and its movement behavior. This remains a largely unresolved aspect of organelle dynamics that can now be addressed by 'multi-parametric' analysis in vivo beyond the capabilities offered by previous approaches.

# Materials and methods

## Key resources table

| Reagent type (species) or resource | Designation | Source or reference | Identifiers | Additional information |
|---|---|---|---|---|
| Biological sample (*Danio rerio*) | Roy | (*Ren et al., 2002*) | mpv17a9/a9 | - |
| Biological sample (*Danio rerio*) | Isl2b:Gal4 | (*Ben Fredj et al., 2010*) | Tg(−17.6isl2b:GAL4-VP16,myl7:EGFP)zc60 | - |
| Genetic reagent (UAS Constructs) | UAS:mitoTagRFP-T; UAS:mitoPAGFP; UAS:memYFP; UAS:mitoDendra2 | (*Köster and Fraser, 2001*) | Identified as in B - there are no specific identifiers | |
| Software | Analysis Software | this paper | https://gitlab.com/groups/3d-spt-orbital-tracking/-/shared | self-written, Matlab 2015b |

## Animals

We used a mutant zebrafish line (Roy) with impaired production of silver pigment (*Ren et al., 2002*), which we crossed with the transgenic driver line Isl2b:Gal4 (kept at a heterozygous background Roy +/-). This line contains a specific promoter that drives expression of the Gal4 transcription factor in zebrafish sensory neurons (*Ben Fredj et al., 2010*). Fish were maintained, mated, and raised as previously described (*Mullins et al., 1994*). Embryos were kept in 0.3x Danieau's solution at 28.5°C and staged as described (*Kimmel et al., 1995*). No randomization or blinding in the selection of zebrafish embryos was used in this study. All experiments with zebrafish larvae were performed according to institutional and government regulations.

## Labeling constructs, screening and mounting

To achieve mosaic labeling, UAS constructs were co-injected (each at a concentration of 10 ng/μl) into fertilized eggs of the Isl2b:Gal4 x Roy fish as described before (*Godinho, 2011*). At 24 hr post-fertilization (hpf), embryos were transferred to 1% N-phenylthiourea (PTU) to inhibit pigmentation. At 2 dpf, embryos were anesthetized using tricaine (at a final concentration of 0.75 mM) and embedded in low melting agarose (0.7–0.8%, Sigma) for screening. Embryos showing a suitable expression pattern (mitoTagRFP-T expressed in isolated sensory neurons in the tail fin) were removed from agarose and allowed to recover in PTU solution overnight. The next day they were again anesthetized and mounted in agarose for imaging. PTU/Tricaine remained present throughout the experiment. When using photosensitive proteins, embryos were maintained in a dark environment and mounted under a dissection microscope with blue light excluding filters. A list of constructs and identifiers are given in the Key resource table.

## Experimental setup

### Microscope design

The instrument used in this work was designed based on a previously described microscope (*Dupont et al., 2013*; *Katayama et al., 2009*). To enhance the response of the system for tracking at high speeds, we replaced the piezo mirror, which had a long response time (>10 ms), with galvanometric mirrors resulting in orbit repetition rates up to 333 Hz. In addition, we replaced the previously used software (SIMFCS, LFD University of California – Irvine, US) with a self-written code written in LabVIEW 2014. The program runs on the host computer and on a deterministic processing unit, consisting of a field-programmable gate array (FPGA) and a real-time processor (cRIO 9082, National Instruments). This upgrade enabled us to achieve real-time control of the system with a jitter of less than 100 µs, implement new modes of operation and to synchronize all hardware components. In the current configuration, the system has a localization precision of 3 nm laterally and ~20 nm axially using 190 nm beads as a calibration sample, depending on the count rate (*Figure 2—figure supplement 2*). Stationary mitochondria in the stem axon were used to measure the localization precision. Using the same calibration procedure, the system achieved an average lateral localization precision of 31.5 nm (21 nm stationary accuracy, 42 nm dynamic accuracy). The lateral localization precision for smaller moving mitochondria (after removing high frequency oscillations by averaging over five localizations) can be estimated through the standard deviations of the orbit displacement histograms and averages 4.6 nm (*Figure 2c*). The accessible tracking area of the microscope is limited by the travel range of the z-objective piezo (100 µm) and the travel range of the sample stage on top of the microscope (>10 cm, using the long-range tracking mode). The tracking algorithm was implemented using a combination of a field programmable gate array (FGPA), a real-time computer and a standard personal computer. This combination allowed us to achieve accurate timing on the submillisecond time scale and a synchronized operation of the microscope as discussed below.

## Tracking software

To achieve accurate timing on the submillisecond time scale, we separated the execution of the tracking code between a host computer (not deterministic) and a deterministic processing unit. The host computer (Intel Core-i7, 16 GB Ram, Solid State Disc for image acquisition) is used to record video frames, to visualize the data, to control the deterministic unit and to receive and save the data. The deterministic unit contains an FPGA and a real-time computer on a single device. The FPGA records incoming TTL pulses, generates the voltages for the galvanometer mirrors and the z-piezo and synchronizes the AOTFs and the EMCCD camera with TTL signals. Furthermore, it sends photon counts, through a direct memory access first-in first-out buffer (DMA-FIFO), to the real-time computer. The communication between the FPGA and the real-time computer occurs instantaneously (with respect to time scales relevant for tracking) as they are mounted on the same device. The communication between the real-time and the host computer occurs over a gigabit Ethernet network, which ensures lossless data transmission. Defining a tracking area on a previously acquired confocal image is the starting point for the tracking algorithm. All necessary orbit properties (orbit size, orbit time, the intensity threshold, starting point of the tracking, etc.) are transferred separately before the experiment or directly before starting the tracking algorithm. The FPGA initiates the orbit around the particle and measures the intensity in 16 sectors of each orbit. After measuring each sector, the photon counts are binned, transferred to the real-time computer and stored in two FIFO buffers, one for each detector. When all 32 elements are available (16 bins × 2 detectors), the buffers are read out and the positioning algorithm calculates the new position of the particle. The calculation is described in detail in the Tracking Algorithm Section. This takes between 100 µs and 700 µs. Depending on whether a particle is detected, which is determined from the current count rate, the positioning algorithm performs two different tasks. If the current count rate is below the user-defined threshold, a searching modus is started. It performs a spiral motion laterally around the starting point until the count rate rises above the threshold, which indicates that a particle has been found. When the count-rate is above the threshold, the position of the particle is determined by the real-time computer. The new position data is written into variables on the FPGA, which are continually read out during the orbit and provide the feedback control of the orbit. Since the new orbit is started directly after the previous one, the first point is biased with the position from the old one. In practice, it is more convenient to choose this 'on the fly' update, which causes no measurable

performance loss, than to wait for the positioning algorithm to finish the calculations. Depending on the chosen settings, the positioning algorithm performs additional tasks, which are explained in detail in the following paragraphs. After the new position was sent to the FPGA, the real-time computer has a certain wait time until the FIFO buffers with the intensity data from a new orbit can be read out again. This time is used to send several data values over the gigabit network to the host computer. These data values include the three-dimensional position, the count rate of both APDs, the time for each orbit, if a particle was tracked during the orbit, the current camera frame and the direction of a recentering event, when long-range tracking is used. When the host computer receives these values, the data is visualized in a user interface and allows the user to follow the particle and to take action, if desired. The algorithm continues to track the particle until the count rate drops below a given threshold. The user can then decide whether she/he wants to continue tracking another particle that diffuses into the search area or stop the tracking experiment. When the user wants to stop the tracking proceedure while the intensity is above threshold, a command is sent over the network to the real-time computer and the FPGA and the data acquisition are stopped.

## Tracking algorithm

The tracking algorithm is implemented in several steps. For the lateral localization of the particle, the signals from both detection channels are summed together. The intensity counts from 16 bins per orbit can be described by a Fourier series:

$$I(\varphi, r) = \frac{a_o(r)}{2} + \sum_{k=1}^{n}(a_k(r)\cos(k\varphi) + b_k(r)\sin(k\varphi))$$

Only the zero- and first-order coefficients are relevant for the localization and are extracted using a Fast Fourier Transformation. The angular position, $\phi$, and the distance to the center of the orbit, $d_r$, are given by:

$$\varphi = arctan\frac{b_1(r)}{a_1(r)} \quad d_r = r_{Orbit} * f(r) * Mod(xy)$$

$$Mod(xy) = \frac{\sqrt{a_1^2(r) + b_1^2(r)}}{a_0(r)}$$

Localization in z is performed through the intensity ratio between the two detection planes. The particle's axial position in relation to the focal plane $d_z$ is given as:

$$d_z = \Delta z_{APD1-APD2} * g(z) * Mod(z) \quad Mod(z) = \frac{I_{APD\,1} - I_{APD\,2}}{I_{APD\,1} + I_{APD\,2}}$$

To minimize the calculation time, both scaling functions, $f(r)$ and $g(z)$, are implemented with a combination of a look-up table and a binary search, which ensures a rapid determination of the new orbit position. The two look-up tables were generated through simulations (Matlab 2012b) in units of the orbit radius ($r_{Orbit}$) and the distance between the two detection planes ($\Delta z_{APD1-APD2}$). Both look up tables are stored on the real-time computer and loaded into the random access memory for reliable and fast accessing during the experiment.

## Simultaneous wide-field imaging

To determine the location of the tracked particle within the specimen, we use a wide-field microscope coupled to the tracking system whose image plane is aligned to the focal plane of the confocal orbital tracking system (*Katayama et al., 2009*). The new tracking software externally triggers each frame of the EMCCD camera and thus allows a temporal synchronization between the tracking coordinates and the wide-field images. Spatial synchronization is achieved through a mapping procedure and subsequent coordinate transformation. A second-order polynomial transformation matrix is created by removing the emission filter and recording the back reflection of the tracking laser at the coverslip for 25 known locations on the EMCCD camera.

## Long-range tracking

One of the advantages of orbital tracking is that it is a feedback approach and can, in theory, follow the particle throughout the whole specimen. In practice, however, as particles travel away from the initial position along the optical axis of the microscope, imperfections in the optics and alignment process lead to a relative shift between the two detection planes. This mismatch between the detection planes results in a loss of localization precision and ultimately to failing of the tracking capability. Furthermore, the field-of-view of the EMCCD camera is limited, in our case to an area of $35 \times 35$ $\mu m^2$. For particles that travel outside of this area, the environmental information is lost. To overcome these difficulties, other groups use a feedback loop to move the sample stage rather than the laser beam (*Welsher and Yang, 2014*; *Lessard et al., 2007*). However, this leads to a decrease in the response time. The approach we have undertaken is to incorporate a second feedback loop into the tracking software that recenters the sample with an additional stage when the particle reaches a predefined distance from the center of the optical axis. Each axis is handled independently. In this way, we keep the fast response of laser scanning for the majority of the trajectory while increasing the accessible tracking area in $x$ and $y$ to several cms. During such a recentering event, the corresponding galvanometer mirror is moved back to the resting position and the tracking algorithm waits until the stage movement is complete. Depending on the axis, this takes $35 \pm 16$ ms (y-axis) to $63 \pm 5$ ms (x-axis). Afterwards the tracking algorithm continues to follow the particle.

## Dark orbits

When the temporal resolution of the microscope is higher than necessary for tracking the particles of interest, photobleaching can be reduced by either decreasing the laser power and the orbital frequency or by turning off the laser during consecutive orbits, a feature coined 'dark orbits'. As the FPGA controls the AOTF as well as the galvanometer mirrors, we can utilize only one out of every $n$th orbit to excite the particle and to determine its location. The laser is turned off for the remaining $n$-1 orbits, thereby effectively reducing the exposure time of the particle and allowing the collection of longer trajectories. When using dark orbits, the bias of the first data point observed when updating 'on the fly' is no longer an issue as there is ample time to center the orbit on the new location of the particle.

## Data collection

### 3D tracking and wide-field imaging

At 3 dpf, selected larvae positive for mitoTagRFP-T were prepared for imaging. Larvae were mounted in low melting agarose in glass-bottom petri dishes. During the experiment, the temperature was maintained at 25° to decrease mechanical drift. Each fish was screened for the expression of mitoPAGFP prior to the experiment by photoconverting a small subset of mitochondria outside the region of interest. Once we identified embryos positive for PAGFP, we activated individual mitochondria with blue light using a $xy$ scanning pattern (405 nm, 80$\mu$W measured before the objective; $34.6 \times 34.6$ $\mu m^2$, that is a region of $256 \times 256$ pixels of $135 \times 135$ $nm^2$ per pixel; 30$\mu$s pixel dwell time). To estimate the photoactivation contrast, we converted five sensory neurons and measured the PAGFP and TagRFP-T channels before and after photoactivation. This resulted in a $\sim$ 25 fold change in ratio (27.8 $\pm$ 2.3; red channel, before-to-after: 0.7 $\pm$ 0.05; green channel: 20.6 $\pm$ 1.5; mean $\pm$sem = 5 cells). We tracked each activated mitochondrion in the green channel (488 nm excitation) with an acquisition speed of 100 Hz (a 5 ms orbit +one 5 ms dark orbit) and simultaneously observed it via the wide-field microscope in the red channel (561 nm excitation, 2 Hz). We stabilized the count rate at approximately 500 photons per orbit (or 100 kHz) by manually adjusting the laser power during the measurement to achieve a constant localization precision of <5 nm in $xy$ and ~30 nm in $z$. The laser power ranged from <1 $\mu$W in the beginning to up to 25 $\mu$W, measured before the objective, at the end of a measurement. The feedback loop for the long-range tracking was activated every time a mitochondrion traveled further than 10 $\mu$m away from the center of the tracking area. Mitochondria were tracked until the intensity counts fell beneath the background threshold level (adjusted individually for each fish). In total, we collected 43 trajectories from 16 individual zebrafish larvae. Trajectories where the algorithm was switching between different mitochondria during the experiment were discarded.

## Repetitive tracking

We investigated the propensity of mitochondria to pause at particular locations along an axon by tracking the motion of several mitochondria along the same region of a neurite. For these experiments, selected larvae positive for mitoDendra2 and mitoTagRFP-T were prepared for imaging at 3 dpf as described above. A stretch of a neurite (approximately 35 µm in length) was repeatedly irradiated using 405 nm light (80µW measured before the objective; $34.6 \times 34.6$ µm$^2$, that is a region of $256 \times 256$ pixels of $135 \times 135$ nm$^2$ per pixel; 30µs pixel dwell time) to convert mitoDendra2 from green to red. Any new mitochondria entering the photoconverted region were tracked (100 Hz, a 5 ms orbit +one 5 ms dark orbit) using 488 nm excitation until they either left the field-of-view or were fully photoconverted from the 488 nm excitation laser to red fluorescence and thus could no longer be tracked in the green channel. After each recorded trajectory, a 405 nm scan was repeated to erase any residual or reemerging fluorescence. Simultaneous wide-field observation was only possible for the first trajectory due to the high rates of mitoTagRFP-T bleaching caused by the 405 nm laser. In total, we recorded 50 trajectories from three neurons (three embryos).

## Tracking precision

### Tracking precision, in vitro

Tracking precision depends on both the signal-to-noise ratio of the measurement as well as the mobility of the particle. We first tested the localization accuracy of stationary particles. 190 nm multi-fluorescent beads (Spherotech) were imbedded in a polymer and tracked using various laser powers. Data are shown in *Figure 1—figure supplement 1*. From the standard deviation of the particle position, we estimate the tracking accuracies to be <3 nm laterally and <21 nm axially for count rates above 200 photons per orbit (or 40 kHz). Next, we tested the precision of tracking of a mobile particle by using a 3-axis piezo stage mounted on the microscope to move the sample with immobilized particles in a sinusoidal pattern over ±2 µm. The particle position was fit to a sinusoidal function and the standard deviation of the residuals used to estimate the tracking precision. For count rates of 1600 photons per orbit (or 320 kHz), no significant degradation of the tracking was observed up to velocities of >25 µm/s laterally and 15 µm/s axially. The slight decrease in accuracy of the *x* axis compared to the *y* axis at velocities above 5 µm/s is a result of the ~0.1 ms delay in updating the position of the particle 'on the fly' at the starting point of the new orbit ($\phi = 0°$).

### Tracking precision, in vivo

To estimate the tracking precision for following mitochondria under in vivo conditions, we tracked immobilized mitochondria in zebrafish embryos. For a count rate of 500 photons per orbit (100 kHz), we measured a localization precision along the minor axis of the mitochondrion of 21 nm (*Figure 1—figure supplement 1e*). When using a piezo stage to move the stationary mitochondria (similarly to the dynamic precision measurements using beads), we determined a tracking precision of 42 nm (*Figure 1—figure supplement 1f*). Moving mitochondria are typically smaller than stationary ones. If we analyzed the standard deviation of the *y* orbit displacement (*Figure 2c*), we achieve an average localization precision of 4.6 nm after removing high-frequency noise by smoothing the trajectory data by five points. Thus, the system is capable of measuring moving mitochondria in vivo with milli-second-and nanometer resolution over substantial parts of axonal arbors.

## Impact of mitochondrial shape

While mitochondria typically have an elliptical shape, which can decrease localization precision in other tracking techniques (*Kao and Verkman, 1994*), our 3D orbital tracking approach is not very sensitive to the shape as the recorded fluorescent signal during each orbit is converted into the frequency domain using a Fourier transformation. This is true as long as the object is on the size of the orbit or smaller. The lateral particle location is determined by the center-of-mass of the fluorescent signal represented in the first- and zero-order Fourier coefficients. Shape information is encoded in higher order Fourier coefficients, which are not used to calculate particle position. When the objects are much larger than the diameter of the orbit and homogenous in labeling, motion of the particle would not lead to a modulation in signal (*Figure 1—figure supplement 3a*). To investigate the influence of shape on orbital tracking, stationary mitochondria were externally moved using a piezo stage and tracked. The imposed trajectories could be completely recovered for mitochondria shorter

than 1.2 µm (*Figure 1—figure supplement 3b*). As moving mitochondria in zebrafish show an average length of 0.71 ± 0.13 µm, we can rule out an influence of the mitochondrial shape on localization precision.

## Data analysis

Data and video analyses were performed using a self-written analysis software program in Matlab 2015b (The Mathworks, Inc).

## Trajectory analysis

To differentiate between active transport and stationary phases in each trajectory, we used a correlation approach. The sample was mounted such that the axons where extended roughly along the *x* axis. The lateral angle, $\varphi$, respective to the *x* axis between two consecutive orbits, was correlated along the trajectory using a sliding window of 64 points leading to a time dependent correlation function given by:

$$Cor(t, \tau) = \frac{1}{(n - \tau)} \sum_{t}^{t+64-\tau} (\varphi_t)(\varphi_{t-\tau})$$

To reduce the dimension of the correlation carpet $Cor(t, \tau)$, we calculated the mean value of the interval $Cor(t, \tau = 0.03 \text{ s})$ to $Cor(t, \tau = 0.06 \text{ s})$ (*Figure 1e*). For a purely active transport process, the correlation amplitude approaches 1. The correlation amplitude decreases for stationary phases. To estimate the correlation amplitude given by stationary phases, we randomized the trajectory. However, as many trajectories have a clear direction of motion, particular angles of *x* occur more frequently and pure randomization of the trajectory is insufficient. Hence, we took the array of angles between consecutive positions in the trajectory and added a copy of the array where the sign of the angles had been reversed. The total array was then randomized and the same correlation analysis was applied. The mean plus five times the standard deviations was used as a threshold level and values above this value were assigned to regions of active transport. The assignment of active and stationary states was then shifted by half the size of the sliding correlation window (64/2 data points) to remove the delay caused by the size of the correlation window. To remove any artifacts introduced by a long-range tracking event, active phases shorter than 150 ms (twice the duration of recentering) are marked as stationary. The trajectory was smoothed after the correlation by five points (50 ms) to reduce high-frequency noise caused by the tracking algorithm. Using simulated data, we tested the analysis method and verified that >90% of the data points are correctly assigned by the analysis.

Each detected active phase was assigned to one of four dynamic populations. To do this, each active phase was first assigned to the retrograde or anterograde direction based on the motion with respect to the cell nucleus. As all fish were roughly aligned along the *x* axis, classification was determined through the direction of the total movement in *x* for each active phase. In the next step, the average velocity from each active phase was determined from the distance traveled and duration of the phase. Two velocity histograms, one for retrograde motion and one for anterograde motion, were then generated from all active phases for an individual trajectory (*Figure 2—figure supplement 1*). As fast motion is active in only one direction for a single mitochondrion trajectory, velocity histograms in the direction of fast motion were fitted with a two component Gaussian distribution using a maximum likelihood approach. Each active phase was then assigned to either the fast transport process (when above a threshold calculated as the mean of the two Gaussian centers) or to the slow state (when below the threshold, *Figure 2—figure supplement 1*).

## Wide-field analysis

We investigated whether the data presented in this study yields results consistent with previously published results (*Plucińska et al., 2012*). A direct comparison was not possible as the measurements were performed at different temperatures and the spatial and temporal resolution of the wide-field data is insufficient for detecting the different transport populations observed using orbital tracking. Thus, we analyzed our simultaneously acquired wide-field images similarly to what was done previously using an ImageJ plugin, MTrackJ (*Meijering et al., 2012*). A comparison of the wide-field data at 25°C with the orbital tracking analysis showed that the wide-field data is an average of the fast, slow and stationary populations (*Table 2*). As expected, a lower average

anterograde and retrograde speed of mitochondria in Rohon-Beard sensory neurons was measured at lower temperatures (25°C compared to 28°C).

To investigate whether mitochondria have a tendency to pause in the vicinity of another mitochondrion, we performed a colocalization analysis using the wide-field images and the orbital tracking data. The positions and lengths of stationary mitochondria were extracted from the simultaneously acquired wide-field video. Moving mitochondria were removed from the video I(x,y,t) using the following equations.

$$\Delta I(x,y,t) = I(x,y,t+10) - I(x,y,t)$$

$$\Delta I(x,y,t) = \begin{cases} ll\Delta I(x,y,t), & if\ \Delta I(x,y,t)<0 \\ 0, & if\ \Delta I(x,y,t)>0 \end{cases}$$

$$I_{stationary}(x,y,t) = I(x,y,t) + I(x,y,t)$$

As the analysis requires a stationary field-of-view, we analyzed regions between recentering events separately. For a video containing $m$ long-range tracking events, $m + 1$ mean images of all the frames between two recentering events were calculated. The mean image after the last long-range event usually contained a very long (>1 min) stationary phase and was discarded. The $m$ mean images were smoothed frame-wise by five pixels in $x$ and $y$ and converted to binary images (threshold: $\mu$ +5*$\sigma$ of each image). To remove artifacts arising from autofluorescence, other mitochondria and other labeled neurites, a mask was applied that excludes all values outside of a 51 × 11 pixel area around the trajectory of the tracked mitochondria. With the built-in Matlab function „regionprobs', we then extract the position and length of each stationary mitochondrion and combined this information with the measured trajectory using the coordinate transformation determined during the calibration (*Figure 3a,b*).

## Statistics

Statistical values were generated by dividing each set of data into 10 equally sized bins and fitting the data either to a Gaussian distribution (*xy* Velocity) or to a single exponential decay (Duration, *xy* Displacement). The average, percentiles, standard deviations and significance levels (two-sided t-test) were then calculated from the center of the Gaussian functions or the decay constants determined through the fit. The statistical values for the *x* orbit displacement were obtained through directly fitting a Gaussian function to each data set.

## Software availability

The tracking and analysis software used in this study is available at https://gitlab.com/groups/3d-spt-orbital-tracking/-/shared (*Wehnekamp, 2015*; copy archived at https://github.com/elifesciences-publications/Orbital-Tracking-Zebrafish2019-).

## Acknowledgements

We thank Leo Hansbauer and Lisa Haddick for help with the data acquisition, Leanne Godinho for help with the image in *Figure 1A* and advice on fish genetics, Kristina Wulliman for zebrafish husbandry, Leanne Godinho, Petar Marinković and Monika Brill for cloning some of the used constructs, Douglas Campbell for help with zebrafish injections and Frank Mieskes for helpful discussions. We acknowledge the late Chi-Bin Chien (U. Utah) for the *Isl2b*:Gal4 line, which was kindly transferred to us by Martin Meyer (King's College London). The authors are grateful for financial support from the Deutsche Forschungsgemeinschaft (DFG) through SFB1032 (Project B3) and SFB870 (Project A11), Priority Program SPP1710, research grants Mi 694/7–1 and 8–1 and the Excellence Clusters Center for Integrated Protein Science Munich (CIPSM), Nanosystems Initiative Munich (NIM) and the Munich Cluster for Systems Neurology (SyNergy). We also thank the Ludwig-Maximilians-Universität, München for support through the Center for NanoScience (CeNS) and the BioImaging Network (BIN). Further support came from the German Center for Neurodegenerative Diseases and the European Research Council under the European Union's Seventh Framework Program (FP/2007–2013; ERC

Grant Agreement n. 616791). G.P. was supported by the Graduate School of the Technical University of Munich (TUM-GS) and a postdoctoral fellowship by the DFG.

## Additional information

### Funding

| Funder | Grant reference number | Author |
|---|---|---|
| Deutsche Forschungsgemeinschaft | SFB1032 (Project B3) | Don C Lamb |
| Fakultät für Chemie und Pharmazie, Ludwig-Maximilians-Universität München | Center for NanoScience (CeNS) and the BioImaging Network (BIN) | Don C Lamb |
| H2020 European Research Council | ERC Grant Agreement n. 616791 | Thomas Misgeld |
| German Center for Neurodegenerative Diseases | | Thomas Misgeld |
| Deutsche Forschungsgemeinschaft | Munich Cluster for Systems Neurology (SyNergy) EXC 2145 | Thomas Misgeld |
| Deutsche Forschungsgemeinschaft | Priority Program SPP1710 | Thomas Misgeld |
| Deutsche Forschungsgemeinschaft | Research grants Mi 694/7+8 | Thomas Misgeld |

The funders had no role in study design, data collection and interpretation, or the decision to submit the work for publication.

### Author contributions

Fabian Wehnekamp, Data curation, Software, Formal analysis, Investigation, Visualization, Writing—original draft, Writing—review and editing; Gabriela Plucińska, Resources, Data curation, Investigation, Writing—review and editing; Rachel Thong, Resources; Thomas Misgeld, Don C Lamb, Conceptualization, Supervision, Project administration, Writing—review and editing

### Author ORCIDs

Thomas Misgeld (iD) https://orcid.org/0000-0001-9875-6794
Don C Lamb (iD) https://orcid.org/0000-0002-0232-1903

### Decision letter and Author response

Decision letter https://doi.org/10.7554/eLife.46059.034
Author response https://doi.org/10.7554/eLife.46059.035

## Additional files

### Supplementary files

• Transparent reporting form
DOI: https://doi.org/10.7554/eLife.46059.022

### Data availability

The analysis software program is available on Gitlab (https://gitlab.com/frmie/Orbital-Tracking-Zebrafish2019; copy archived at https://github.com/elifesciences-publications/Orbital-Tracking-Zebrafish2019-) and the wide-field images and trajectories are available on Zenodo. Source data files have been provided for all the figures.

The following datasets were generated:

| | **Database and** |
|---|---|

| Author(s) | Year | Dataset title | Dataset URL | Identifier |
|---|---|---|---|---|
| Wehnekamp F, Plucińska G, Thong R, Misgeld T, Lamb DC | 2019 | Wide-field Images and Trajectories | http://dx.doi.org/10.5281/zenodo.2813946 | Zenodo, 10.5281/zenodo.2813946 |
| Wehnekamp F, Plucińska G, Thong R, Misgeld T, Lamb DC | 2019 | Wide-field Images and Trajectories | http://dx.doi.org/10.5281/zenodo.2815430 | Zenodo, 10.5281/zenodo.2815430 |
| Wehnekamp F, Plucińska G, Thong R, Misgeld T, Lamb DC | 2019 | Wide-field Images and Trajectories | http://dx.doi.org/10.5281/zenodo.2815550 | Zenodo, 10.5281/zenodo.281550 |
| Wehnekamp F, Plucińska G, Thong R, Misgeld T, Lamb DC | 2019 | Wide-field Images and Trajectories | http://dx.doi.org/10.5281/zenodo.2815703 | Zenodo, 10.5281/zenodo.2815703 |
| Wehnekamp F, Plucińska G, Thong R, Misgeld T, Lamb DC | 2019 | Wide-field Images and Trajectories | http://dx.doi.org/10.5281/zenodo.2815801 | Zenodo, 10.5281/zenodo.2815801 |

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
