## [Decision Letter]

Thank you for submitting your article "Nanoresolution real-time 3D orbital tracking for studying mitochondrial trafficking in vertebrate axons in vivo" for consideration by *eLife*. Your article has been reviewed by three peer reviewers, one of whom is a member of our Board of Reviewing Editors, and the evaluation has been overseen by Didier Stainier as the Senior Editor. The following individual involved in review of your submission has agreed to reveal their identity: Subhojit Roy (Reviewer #3).

The reviewers have discussed the reviews with one another and the Reviewing Editor has drafted this decision to help you prepare a revised submission.

There was a consensus among the reviewers that the work represented an important technical advance and indeed is likely the most high resolution and detailed description of axonal mitochondrial transport in vivo to date. As such, the manuscript is very well-suited, in principle, for publication in the Tools and Resources category which publishes significant technological or methodological advances. There was also a consensus that, for consideration as a standard article, the biological aspects of the paper would need to be expanded. This might consist of an analysis of changes in motility during development or regeneration, comparing peripheral and central axons, trunk axons vs. higher-order branches or termini, analysis of mutants, comparison to transport of other organelles, etc. For the Tools and Resources category, however, that will not be necessary. Some changes to the manuscript will, however, be necessary for it to be acceptable, as described below.

Summary:

The manuscript describes an in vivo imaging system in larval zebrafish sensory neurons that allows the authors to track mitochondria with remarkable temporal and spatial resolution (XY < 5 nm, Z < 30 nm, 100 Hz). An automated system that re-centers the stage when an organelle reaches the edge of field-view also allows them to track a single mitochondrion for long distances (> 100 um). The authors find a previously unknown 'slower state' of transport of mitochondria that corresponds with directional changes, and provide a detailed description of the changes between fast, slow, and stationary states, for both anterograde and retrograde movement. The authors have provided comprehensive explanations of the hardware arrangement, and have conscientiously calibrated the imaging capabilities of the system. The automatic re-centering feature of the system is useful, since it allows tracking of objects at high magnification over long distances. The increased temporal and spatial resolution of the system is likely not necessary for many studies of axonal trafficking, but might provide unanticipated insights in some cases. The paper is very well written and a joy to read.

Essential revisions:

1) More anatomical context for the images and videos is needed. The only panel that shows the larger neuronal context for the experiments (as opposed to quantitative analysis) is Figure 1B. It would be helpful to see more zoomed-out widefield images to place the tracks and analyses into context.

2) More images of the photoactivation and photoconversion experiments, since other scientists who might want to use these systems should able to assess how well they worked. From the images in Figure 1B, it looks like there is a lot of background photoactivation and, as a result, photoactivation does not appear to provide much improvement in contrast.

3) The authors are entitled to whatever interpretation they wish to make in the Discussion. However, I would encourage them to consider a few interpretations of the slow movement states that do not involve novel behaviors of the kinesin or dynein motors and that do not invoke an actin myosin mechanism. The latter is unattractive because there is very little evidence (none in neurons) in support of myosin-based movement of mitochondria (as opposed to anchoring by myosin) and little reason to think that an appropriately oriented actin fiber would be immediately at hand for a smooth hand-off from the kinesin or dynein motor. On the other hand, movement mediated by the well-documented kinesin and dynein motors might achieve slower progress in a highly resistive environment though operating in their normal mode. The authors should also discuss the potential of shape changes at these constrictions. The mitochondria are certainly not rigid bodies and are likely to be moved by multiple motors that are not necessarily uniform along their surface. Thus if the leading edge of the organelle is forced to pause by an obstruction but the trailing edge moves forward, the rate of progress of the centroid would appear to be in a slower state. A similar distortion could occur if the leading edge is free of obstruction while the rest of the mitochondria remains immobilized by an obstruction. How would this appear in the orbital tracking? The discussion of mitochondrial shape in Figure 1—figure supplement 3 does not consider shape changes, only shape differences. Unless the reviewers have overlooked some technical feature of the imaging, these appear far more likely explanations of the slow states. The authors are urged, at a minimum, to include the arguments for and against these possibilities if there are arguments to be made.

4) The authors are strongly discouraged throughout from using the term "passive" to describe the periods with no net progress. Stationary or static seem more apt. Passive implies that motors are not active, but this state could as easily be explained by motors that are thwarted by an anchor or obstacle or by a tug of war of opposing motors.

5) Many readers will be accustomed to analysis by kymography. In kymographs it is not uncommon to encounter mitochondria that have no net movement for the duration of imaging and yet are not stationary, but jiggle back and forth in place over short distances. It might be helpful to inform the reader if any such activity was observed in this study and whether it is equivalent to or completely different from the slow or passive states that interrupt the progressive movements described here.

6) The authors are encouraged, in the Discussion, to place their findings in the bigger context of fast transport and compare the rates measured here to those detected in kymography, by in vitro assays of motor speeds, and relative to the older literature on pulse-chase radiolabeling experiments. To what extend to the slower phases described here influence measurements made elsewhere?

---

## [Author Response]

Essential revisions:1) More anatomical context for the images and videos is needed. The only panel that shows the larger neuronal context for the experiments (as opposed to quantitative analysis) is Figure 1B. It would be helpful to see more zoomed-out widefield images to place the tracks and analyses into context.

We have addressed this shortcoming by now adding a new panel to Figure 1 (new 1A), that shows a transmission image of the zebrafish tail with a typical Rohon-Beard neuron labeled by a membrane-targeted fluorescent protein. We indicate a typical recording position thereby giving the anatomical context for the figure panels below, where mitochondrial imaging is shown and it is a bit harder to contextualize for the non-zebrafish reader.

2) More images of the photoactivation and photoconversion experiments, since other scientists who might want to use these systems should able to assess how well they worked. From the images in Figure 1B, it looks like there is a lot of background photoactivation and, as a result, photoactivation does not appear to provide much improvement in contrast.

Thank you for noticing this flaw in our presentation – due to the contrast inversion of this insert, the images presented in the original submission did not represent a proper linear look-up table. We have now remedied this, by providing the images on their normal dark-field background (revised inset in Figure 1C, originally 1B) and by also showing the photo-converted channel (green) separately. As is apparent, the background is actually relatively low. We have also performed some measurements of the photoconversion contrast in some mitoPA-GFP/RFP labeled Rohon-Beard neurons in vivo and find a conversion contrast of about 25-fold (27.8 ± 2.3; red channel, before-to-after: 0.7 ± 0.05; green channel: 20.6 ± 1.5; mean ± sem = 5 cells). These results are now incorporated in the Materials and methods section (subsection “3D tracking and wide-field imaging”). Finally, it is worth noting that the orbital tracking is relatively insensitive to local background that is, e.g. caused by off-target photoconversion by scattered light, as the orbit moves away from the photoconversion site with the tracked object into areas that have not been exposed to the photoconversion wavelength. Thus, if the tracking successfully picks up an object in the beginning, the signal-to-background of this object improves as soon as it move into a pristine environment.

3) The authors are entitled to whatever interpretation they wish to make in the Discussion. […] The authors are urged, at a minimum, to include the arguments for and against these possibilities if there are arguments to be made.

While we appreciate the freedom to make fools of ourselves in the Discussion, we agree with this comment and have now expanded the Discussion section to broaden the considered mechanisms to include the suggested alternatives (such as shape changes), and acknowledges the scant evidence for an involvement of myosin in mitochondrial movements in neurons. We still maintain the view that the directional symmetry of the slow movement behavior suggests a uniform motor or motor/anchor combination as the underlying force, but we fully acknowledge that, without a molecular exploration of the underlying mechanisms, any such preference remains a speculation and should be labeled as such.

4) The authors are strongly discouraged throughout from using the term "passive" to describe the periods with no net progress. Stationary or static seem more apt. Passive implies that motors are not active, but this state could as easily be explained by motors that are thwarted by an anchor or obstacle or by a tug of war of opposing motors.

We agree and have revised ‘passive’ to ‘stationary’ as this is the most descriptive term. As discussed below (point 5) and recently further demonstrated in Gutnick et al., 2019, there are multiple ‘stationary’ states that will likely be differentiated in the future by molecular or kinetic characteristics. As pointed out below, our approach might aid in the latter classification, but we have not thus far performed in depth analysis of ‘stationary’ states with orbital tracking.

5) Many readers will be accustomed to analysis by kymography. In kymographs it is not uncommon to encounter mitochondria that have no net movement for the duration of imaging and yet are not stationary, but jiggle back and forth in place over short distances. It might be helpful to inform the reader if any such activity was observed in this study and whether it is equivalent to or completely different from the slow or passive states that interrupt the progressive movements described here.

The reviewers are making an interesting point. As we have not tracked stationary mitochondria systematically, we cannot really place this behavior into our movement schemes. We have inserted a statement in the text making this clear, and point out that different stationary states exist (citing Gutnick at al., 2019), which we, however, did not try to categorize.

6) The authors are encouraged, in the Discussion, to place their findings in the bigger context of fast transport and compare the rates measured here to those detected in kymography, by in vitro assays of motor speeds, and relative to the older literature on pulse-chase radiolabeling experiments. To what extend to the slower phases described here influence measurements made elsewhere?

We thank the reviewers for making this interesting point and address it now in the Discussion, where we provide a short paragraph on the speeds that we measured with those of previous single-particle measurements and the known behavior of reconstituted motors, as well as the bulk behavior deduced from pulse-chase experiments.